# *Toxoplasma gondii* at the Host Interface: Immune Modulation and Translational Strategies for Infection Control

**DOI:** 10.3390/vaccines13080819

**Published:** 2025-07-31

**Authors:** Billy J. Erazo Flores, Laura J. Knoll

**Affiliations:** Department of Medical Microbiology and Immunology, University of Wisconsin-Madison, Madison, WI 53706, USA; berazo@wisc.edu

**Keywords:** *Toxoplasma gondii*, host response, immune modulation, programmed cell death, therapies, vaccines

## Abstract

*Toxoplasma gondii* is an intracellular protozoan found worldwide that is capable of infecting nearly all warm-blooded animals, including humans. Its parasitic success lies in its capacity to create chronic infections while avoiding immune detection, altering host immune responses, and disrupting programmed cell death pathways. This review examines the complex relationship between *T. gondii* and host immunity, focusing on how the parasite influences innate and adaptive immune responses to survive in immune-privileged tissues. We present recent findings on the immune modulation specific to various parasite strains, the immunopathology caused by imbalanced inflammation, and how the parasite undermines host cell death mechanisms such as apoptosis, necroptosis, and pyroptosis. These immune evasion tactics enable prolonged intracellular survival and pose significant challenges for treatment and vaccine development. We also review advancements in therapeutic strategies, including host-directed approaches, nanoparticle drug delivery, and CRISPR-based technologies, along with progress in vaccine development from subunit and DNA vaccines to live-attenuated candidates. This review emphasizes the importance of *T. gondii* as a model for chronic infections and points out potential avenues for developing innovative therapies and vaccines aimed at toxoplasmosis and similar intracellular pathogens.

## 1. Introduction

*Toxoplasma gondii* is an obligate intracellular protozoan parasite and the causative agent of toxoplasmosis, a potentially deadly disease [1]. It is estimated that approximately 30% of the global human population is infected with this parasite [2]. In most healthy individuals, *T. gondii* infection is either asymptomatic or presents with mild flu-like symptoms. The host immune system can typically control the infection damage if the strain encountered is not highly virulent and the host is immunocompetent. In cases of congenital infection or immunocompromised individuals, toxoplasmosis can cause severe complications, including fatal encephalitis, ocular damage, and stillbirth [3,4,5]. The early symptomatic phase of the infection, known as the acute stage, is characterized by the rapid replication of the parasite tachyzoite form. During this stage, tachyzoites disseminate throughout the host body. In reaction to the host immune response and other factors that remain incompletely understood, tachyzoites transition into the slow-growing encysted stage of the parasite, called bradyzoites [6]. This transition marks the onset of the chronic infection. In vitro, the conversion of tachyzoites to bradyzoites has been observed under high pH conditions, exposure to sodium arsenite, and temperature changes [7,8]. However, the precise mechanisms regulating this conversion in vivo are not yet fully characterized. Although *T. gondii* is capable of infecting nearly all nucleated cells [9], bradyzoite cysts are primarily found in mature skeletal muscle and brain neurons, potentially remaining in this area indefinitely and undetected [10]. With a weakened immune system, reactivation of the bradyzoite cysts to tachyzoites can occur [6] leading to severe pathology, such as encephalitis. *T. gondii* infection has been connected to various neurological effects in hosts, especially during the chronic stage. In both animal studies and humans, long-lasting bradyzoite cysts in neural tissue are associated with neuroinflammation [11], changes in neurotransmitter regulation [12,13], and behavioral alterations [12,14]. These findings raise concerns about the parasite’s possible influence on host cognition and neuropsychiatric disorders.

The host immune response against *T. gondii* is a complex, multi-layered defense system involving both innate and adaptive immunity. During the early stages of infection, the innate immune system plays a critical role in detecting and responding to the parasite. While it is critical for the host to develop a robust but balanced innate immune response, *T. gondii* has evolved sophisticated strategies to manipulate host defense pathways, allowing it to persist within host cells. This delicate interplay between host defense and parasite survival determines the outcome of the infection.

This review aims to provide a comprehensive overview of the host immune responses to *T. gondii* infection, emphasizing the delicate balance between protective immunity and immune evasion strategies employed by the parasite. It will explore the host’s innate and adaptive immune mechanisms, including cytokine signaling, cellular immune responses, and the activation of various immune pathways. Finally, the review will discuss the implications of these interactions for host–pathogen dynamics and potential therapeutic strategies.

### Systematic Literature Search Methodology

To ensure a thorough coverage of relevant studies for this review, a systematic search was conducted across multiple scientific databases. Searches were performed in PubMed, Web of Science, and Google Scholar, covering articles published up to 2024. Keywords incorporated various combinations such as “*Toxoplasma gondii*,” “immune response,” “programmed cell death,” “host–pathogen interaction,” “immune evasion,” “vaccine development,” “host-directed therapy,” and “parasite persistence.” Boolean operators (AND/OR) were used to refine and include relevant intersections of topics.

The inclusion criteria encompassed original research articles, reviews, and preclinical or clinical studies focusing on *T. gondii* biology, host immune responses (both innate and adaptive), immune evasion mechanisms, programmed cell death, and therapeutic or vaccine strategies. Preference was given to peer-reviewed publications in English and available as full texts. Studies unrelated to *T. gondii*, lacking sufficient mechanistic detail, or non-peer-reviewed content were excluded. Additional relevant papers were found through manual screening of reference lists from key articles. In total, 150 articles were thoroughly reviewed, with 121 included in the final analysis. The selected literature was critically assessed for relevance, methodological rigor, and its contribution to current understanding. The articles included are cited, with an average per article of 236.52, and an H-index of 73.

To resolve conflicting results across various experimental models, including murine and human cell lines, and in vivo systems, a comparative evidence synthesis approach was employed. This method integrates data considering experimental context, strain differences, and immune responses. The emphasis was on ensuring mechanistic consistency and reproducibility across models to identify broadly supported conclusions, while also acknowledging model-specific limitations.

## 2. Life Cycle of *T. gondii*

*T. gondii* has a complex life cycle that includes both asexual and sexual stages. The asexual cycle occurs in all warm-blooded animals, including humans, where the parasite exists in two main forms: tachyzoites and bradyzoites [15,16]. In contrast, the sexual cycle is restricted to the feline intestine, the only environment in which *T. gondii* undergoes gametogenesis and oocyst formation. It was recently discovered that the sexual development of *T. gondii* is triggered by high levels of linoleic acid [17]. Linoleic acid accumulates due to the feline-specific lack of intestinal delta-6-desaturase activity, the rate limiting step in the conversion of linoleic acid to arachidonic acid. It was also recently observed that inhibition of *T. gondii* HDAC3 (*Tg*HDAC3) hyperacetylates *T. gondii* histones and induces the expression of sexual stage genes [18]. The *T. gondii* transcription factors AP2XII-1 and AP2XI-2 bind DNA as heterodimers at merozoite promoters and recruit MORC and HDAC3, thereby limiting chromatin accessibility and transcription of early sexual stage genes [19]. The fact that inhibition of *Tg*HDAC3 initiates the *T. gondii* sexual cycle suggests the presence of HDAC inhibitor (s) unique to the feline gut. Felines primarily become infected by ingesting tissue cysts containing bradyzoites from prey animals that have acquired toxoplasmosis. Cats can be infected with oocysts, but the prepatent period is longer compared to tissue cysts [20]. Within the feline intestinal epithelium, the parasite differentiates into male and female gametes, which fuse to form oocysts that are subsequently shed in the feces [21,22]. Oocysts undergo sporulation in the environment to produce infectious sporozoites, which can infect new intermediate hosts and thereby complete the life cycle [23].

### 2.1. Routes of Infection

The primary routes of human infection include ingesting contaminated food or water, zoonotic transmission, and congenital transmission [23]. The most common route is the oral ingestion of tissue cysts or oocysts [24]. Humans can become infected by consuming undercooked or raw meat that contains bradyzoite cysts. In addition, ingesting water, fruits, or vegetables contaminated with sporulated oocysts shed by infected cats represents a significant transmission pathway and may lead to more severe disease than infections derived from tissue cysts [25].

Once ingested, the cyst wall is dissolved by proteolytic enzymes found in the stomach and small intestine [7]. Tachyzoites are released in the gastrointestinal tract to infect cells and travel through the bloodstream to reach tissues such as the brain, eyes, and muscles. Congenital transmission occurs when a pregnant woman acquires a primary infection during pregnancy. Tachyzoites can cross the placental barrier, leading to congenital toxoplasmosis, which may cause severe complications such as neurological impairment, ocular lesions, or stillbirth [26]. The risk of transmission and the severity of the disease largely depend on the gestational age at the time of infection. Maternal infections during the first trimester have reduced fetal transmission rates, but are associated with more severe outcomes than those occurring later in pregnancy, with high levels of fetal transmission [27]. Direct contact with infected cat feces is a less common but important transmission route. Cats, as the definitive hosts, shed oocysts in their feces after ingesting infected prey. Humans may inadvertently ingest oocysts through improper handling of cat litter or contact with food grown in contaminated soil, particularly uncooked fruits and vegetables. Understanding these transmission routes is critical for effective prevention, especially among high-risk populations such as pregnant women and immunocompromised individuals. Preventive measures, including thoroughly cooking meat, washing fruits and vegetables, and practicing proper hygiene when handling cat litter or soil, can significantly reduce the risk of infection.

### 2.2. Acute and Chronic Infection

*T. gondii* infection progresses through two main stages: acute and chronic. In humans and mice, acute infection is characterized by the rapid proliferation and spread of the tachyzoite form. After oral ingestion of tissue cysts or oocysts, bradyzoites and sporozoites differentiate into tachyzoites that disseminate systemically through the bloodstream by the circulation of tachyzoite-infected leukocytes in the blood [28,29]. This stage often triggers a strong innate immune response, characterized by elevated levels of IL-12 and IFN-γ, which are critical for limiting parasite replication [30].

After the parasite has spread throughout the body, reaching areas such as the brain and muscle tissue, immune pressure drives the transition from tachyzoites to bradyzoites [6,31]. This slow-replicating form resides within long-lived tissue cysts, marking the onset of chronic infection. In humans, chronic toxoplasmosis is usually asymptomatic but can reactivate in immunosuppressed individuals, leading to severe conditions like encephalitis [4,32]. The ability of *T. gondii* to establish a lifelong chronic infection reflects its evolved strategies to evade immune clearance and highlights the challenges in therapeutic eradication.

## 3. Host Immune Response to *T. gondii*

*T. gondii* has evolved a complex life cycle and sophisticated strategies that enable it to thrive within host cells, evade immune detection, and manipulate host cellular processes for prolonged survival. A defining characteristic of *T. gondii* is its ability to infect nearly all nucleated cells and replicate rapidly within them [33]. Once inside a host cell, the parasite reprograms host functions to create a protective intracellular niche, exploit host resources, and evade immune clearance [34,35,36,37]. It can also disrupt critical host defense mechanisms, including programmed cell death (PCD) [38,39,40]. Understanding how the host immune system responds to *T. gondii* is crucial, as it determines the outcome of the infection and provides insight into potential therapeutic targets for limiting parasite replication and preventing disease progression.

### 3.1. Innate Immune Response

Survival during toxoplasmosis relies on a robust and balanced innate immune response. As the first line of defense, immune cells in the intestinal epithelium, including innate lymphoid cells, are among the initial responders to the parasite during oral infection [41,42,43]. These cells detect the presence of the parasite and initiate early immune responses to prevent its spread. Detection is facilitated by pattern recognition receptors, such as Toll-like receptors (TLRs), which identify pathogen-associated molecular patterns in the parasite [43,44,45].

Mice possess 13 TLRs, while humans express 10 [46]. In mice, TLR11 and TLR12 identify *T. gondii* profilin [47,48], triggering a MyD88-dependent signaling cascade [48,49] and the subsequent production of IL-12 by CD8α^+^ and CD8α^−^ dendritic cells [50,51,52,53]. IL-12 then stimulates natural killer cells and T cells to produce IFN-γ [54,55], a critical cytokine for controlling the parasite. In humans, TLR11 and TLR12 are non-functional [56]. Instead, detection is likely mediated through other TLRs, such as TLR3, TLR7, TLR8, and TLR9, which also signal through MyD88 to induce IL-12 [43,56].

MyD88, IL-12, and IFN-γ are critical for resistance to *T. gondii*; mice lacking any of these components quickly succumb to infection [52,57,58]. IFN-γ plays a central role in activating antimicrobial pathways, including the induction of immunity-related GTPases (IRGs) and guanylate-binding proteins (GBPs) [59,60], which disrupt the parasitophorous vacuole and compromise parasite survival [61,62]. Furthermore, nitric oxide (NO) produced by activated macrophages aids in parasite killing, although excessive NO production can lead to immunopathology [43,63,64].

### 3.2. Adaptive Immune Response

The adaptive immune response to *T. gondii* is initiated and shaped by early proinflammatory cytokines, notably IL-12 [65]. This response encompasses both B and T lymphocytes. Th1-polarized CD4^+^ T cells play a key role in the secretion of IFN-γ [66], which enhances macrophage activation [67,68], promotes antigen presentation [30], and stimulates the creation of reactive oxygen and nitrogen species to eradicate intracellular parasites [69,70]. Similarly, CD8^+^ T cells release IFN-γ [71] and eliminate *T. gondii*-infected cells via perforin and granzymes [72,73]. This cytotoxic function is crucial for eliminating the parasite when it is already present intracellularly. B cells assist the adaptive response by generating antigen-specific antibodies. During the early stage of acute infection, IgM antibodies are produced, whereas IgG antibodies are created later to provide long-term protection. These antibodies neutralize extracellular tachyzoites, hinder additional invasion of host cells, and are used as markers of exposure [1].

## 4. Immunopathology: Balancing Inflammation and Regulation

In both mice and humans, *T. gondii* elicits a robust proinflammatory Th1-polarized immune response that is pivotal for controlling the infection [65,74]. In immunocompetent individuals, this response usually eliminates most parasites, with any remaining ones confined to the tissue cysts, resulting in an asymptomatic or mild infection. However, failing to mount an effective proinflammatory response leads to uncontrolled parasite replication and severe disease manifestations such as toxoplasmic encephalitis [75].

While inflammation is essential to restrain parasite replication, an excessively vigorous immune response can cause collateral tissue damage [76,77]. Common sites of immunopathology include the brain and the eye, where intense inflammation can harm host tissues. For instance, oral high-dose infection of mice with *T. gondii* can result in the overproduction of inflammatory cytokines, leading to severe damage in the small intestine and necrotizing ileitis [78]. Similarly, immune-driven inflammation can damage retinal tissues in the eye, contributing to vision loss due to ocular toxoplasmosis [79]. These examples highlight the need for a balanced immune response that effectively eliminates the parasite while minimizing tissue injury.

To counteract immunopathology, the host employs anti-inflammatory mechanisms. The cytokines IL-10 and TGF-β and regulatory T cells (Tregs) are key players in this regulation. Tregs secrete IL-10 and TGF-β to suppress inflammatory mediators such as IFN-γ and IL-12 to reduce the activity of overactive effector T cells [80,81,82]. IL-10 plays a particularly crucial role in regulating Th1 responses; mouse models lacking IL-10 exhibit elevated levels of IFN-γ and IL-12, leading to lethal inflammation and necrotic liver lesions, even when parasite burdens are controlled [83]. Increased IL-10 or Treg activity can reduce inflammation, as seen in acute ocular toxoplasmosis models, where IL-10 regulates inflammation [84]. TGF-β, produced by Tregs and resident tissue cells, also helps suppress local inflammation and protects sensitive tissues such as neurons and retinal cells [85,86].

*T. gondii* can exploit these anti-inflammatory pathways to enhance its persistence. Parasite molecules, including microneme proteins MIC1 and MIC4, have been shown to induce IL-10 production in host cells [87], promoting a permissive environment that limits immune activation. By boosting IL-10 expression and expanding Treg responses, the parasite tempers host immunity to avoid elimination. This immunomodulatory strategy enables *T. gondii* to establish long-term chronic infections, particularly in immune-privileged tissues like the brain and eye, often with limited clinical symptoms. Therefore, the dynamic balance between inflammation and regulation is central to the pathogenesis of toxoplasmosis: a properly calibrated immune response controls the infection while minimizing damage, whereas an imbalance in either direction can lead to uncontrolled disease or tissue injury.

## 5. Strain-Specific Immune Responses

The host immune response to *T. gondii* is significantly affected by the genetic background of the parasite. The three primary clonal lineages—Type I, Type II, and Type III—demonstrate distinct profiles of virulence, immune modulation abilities, and disease outcomes in experimental models and natural infections [88,89]. Type I strains are highly virulent in mice and are characterized by rapid replication and a low lethal dose [90]. These strains suppress host immunity by secreting virulence factors such as ROP18 and ROP5, which antagonize IFN-γ induced IRGs [90,91]. As a result, the host’s innate and adaptive immune responses are quickly overwhelmed, often resulting in acute mortality [92]. Type II strains are less virulent but more immunogenic. They elicit robust IL-12 and IFN-γ responses, which are essential for controlling the parasite. These strains are commonly used in immunological studies due to their ability to establish chronic infections marked by tissue cysts in the brain and muscles. However, the strong inflammatory responses they provoke can contribute to immunopathology, as seen in ocular and cerebral toxoplasmosis models. In mice, type III strains are considered avirulent and induce a comparatively weak immune response. They stimulate lower levels of proinflammatory cytokines and minimal immune activation.

Beyond the canonical three clonal lineages, atypical and recombinant strains are common in South America [93], and exhibit a broad spectrum of virulence and immunomodulatory profiles. Some of these strains show high replication capacity and effective immune evasion mechanisms, presenting a greater risk of severe disease even in immunocompetent hosts. Ultimately, the outcome of infection, whether asymptomatic persistence or severe disease, is influenced by the complex interaction between the parasite genotype and the host’s immune competence. Understanding strain-specific immune responses is crucial for improving vaccine development, therapeutic interventions, and accurate risk assessment across different populations.

## 6. Programmed Cell Death in Host Defense

PCD plays a critical role in the host’s defense against intracellular pathogens such as *T. gondii*. By eliminating infected cells, PCD removes the intracellular niche, deprives the pathogen of access to host resources, and limits further spread. However, the type of cell death pathway activated, whether apoptosis, necroptosis, or another form, depends on various factors, including the pathogen’s characteristics, infectious burden, tissue environment, and the host’s immune status. Notably, even the same pathogen may trigger different forms of PCD based on the strain type and stage of infection, adding further complexity to the host’s response. This immune decision becomes even more challenging because apicomplexan parasites like *T. gondii* have developed sophisticated strategies to interfere with or subvert host PCD pathways for their benefit. Consequently, while PCD is essential for pathogen control, it also poses potential risks, including damage to surrounding host tissues due to excessive inflammation and immune cell recruitment [94].

Apoptosis is a non-inflammatory form of PCD that is characterized by the controlled fragmentation of cellular contents into membrane-bound apoptotic bodies, which are then phagocytosed without inducing inflammation. This containment is particularly beneficial in immune-privileged areas, such as the brain, where limiting immune-mediated damage is crucial. In the context of toxoplasmosis, apoptosis may aid in reducing neuroinflammation and protecting against severe pathology [95], especially in immunocompromised individuals. *T. gondii* can inhibit or delay apoptotic pathways, enabling continued intracellular replication. In contrast, necroptosis is a proinflammatory form of PCD that involves membrane rupture and the release of intracellular contents, which act as damage-associated molecular patterns, leading to the recruitment of immune cells and an amplification of inflammatory responses. This process can enhance pathogen clearance by exposing the parasite to immune effectors in the extracellular environment. However, if not tightly regulated, necroptosis can cause excessive tissue damage and contribute to immunopathology.

Thus, the host faces a strategic dilemma: inducing apoptosis may limit tissue damage but allow for parasite survival, while triggering necroptosis may enhance immune clearance at the expense of harming host tissues. The choice between these pathways reflects the ongoing evolutionary arms race between host defenses and parasite survival strategies and highlights the complexity of host–pathogen interactions.

## 7. Subversion of Host Cell Death and Immune Pathways by *T. gondii*

*T. gondii* has evolved various strategies to manipulate host cell death pathways and immune responses, thereby promoting its survival, replication, and long-term persistence. Apoptosis is significantly inhibited in *T. gondii*-infected cells. The parasite interferes with the intrinsic (mitochondrial) and extrinsic (death receptor) apoptotic pathways through various mechanisms, including the prevention of cytochrome c release, suppression of caspase activation, modulation of mitochondrial integrity, apoptosome assembly, and signaling intermediates [37]. Infected cells become resistant to normally lethal stimuli such as Fas ligand [96,97], growth factor withdrawal [98], and cytotoxic granzyme B [99] as well as the activation of pro-survival pathways like NF-κB [100]. By preventing premature host cell death, *T. gondii* preserves its intracellular niche and maintains an optimal environment for replication.

The inhibition of caspase-8 and death receptor signaling also impairs one arm of necroptosis. Nevertheless, under certain conditions, such as strong TNF-α or IFN-γ signaling, the activation of the RIPK3 pathway can still drive necroptosis in the absence of apoptosis. To avoid this, *T. gondii* appears to restrict the upstream signals necessary for necroptotic activation. The parasite also actively limits pyroptosis, a rapid, inflammasome-dependent form of cell death. In hosts with functional inflammasome sensors (e.g., NLRP1, NLRP3), macrophages may detect *T. gondii* and initiate pyroptosis [101]. However, in cases of regulating pyroptosis, *T. gondii* can potentially maintain the integrity of its parasitophorous vacuole, protecting its components from detection in the cytosol and thereby preventing inflammasome activation. By modulating apoptosis, necroptosis, and pyroptosis, *T. gondii* ensures the survival of its host cell until the parasite is ready to egress and infect new cells.

Beyond interfering with host cell death, *T. gondii* modulates immune signaling and antigen presentation to prevent immune clearance [102,103,104]. To counter the Th-1 type immune response needed for host control of the parasite, *T. gondii* deploys a range of secreted effectors. For instance, the dense granule protein GRA15 (expressed in Type II strains) activates NF-κB in host cells, enhancing proinflammatory cytokine production, including IL-12 [105]. In contrast, more virulent Type I strains expressing ROP18 can potentially suppress NF-κB signaling, thereby reducing IL-12 output and hindering early immune activation [106]. Also, the secreted effector TgIST traffics to the host nucleus, disrupting STAT1-dependent gene expression, a central aspect of IFN-γ signaling [107]. *T. gondii* infection also downregulates MHC class II expression on antigen-presenting cells, primarily by blocking STAT1 nuclear translocation and transcriptional activity [107]. This suppression of antigen presentation aids the parasite in evading detection by CD4^+^ T cells. Simultaneously, the parasite promotes host-derived immunosuppressive molecules, such as IL-10 and SOCS proteins [108], which further dampen Th1 responses and prevent complete immune activation.

*T. gondii* also directly interferes with intracellular immune effectors. In IFN-γ stimulated cells, particularly in mice, GTPases such as IRGs and GBPs target the parasitophorous vacuole, leading to its rupture and the destruction of the parasite. To counteract this, virulent strains of *T. gondii* express a set of rhoptry proteins, including ROP18, ROP17, ROP5, and GRA7, that phosphorylate and inactivate IRGs, preventing their oligomerization and targeting of the vacuole [109]. This action effectively turns off a major innate defense mechanism. A similar strategy applies to GBPs, whose recruitment to the vacuole is also hindered by parasite virulence factors. Although GBPs are IFN-inducible in mice and humans, their precise interactions with *T. gondii* effectors are still being unraveled. By blocking the recruitment and activation of IRGs and GBPs, *T. gondii* avoids vacuole lysis, exposure of its antigens, and subsequent inflammasome activation. This strategy allows the parasite to remain hidden and protected within its host cell.

Through the coordinated action of rhoptry and dense granule effectors like ROP18, GRA15, and TgIST, *T. gondii* manipulates both cell-intrinsic and immune signaling pathways. These mechanisms enable the parasite to evade immune clearance, replicate undetected, and establish chronic infections that may persist for the host’s lifetime. This highly evolved subversion of host defenses highlights *T. gondii* success as a widespread and enduring intracellular pathogen.

## 8. Therapeutic and Vaccine Strategies

Current treatments for toxoplasmosis primarily depend on antifolate drug combinations targeting the actively replicating tachyzoite stage of the parasite. The most common regimen includes pyrimethamine and sulfadiazine, supplemented with folinic acid to reduce bone marrow toxicity [110]. Alternatives like trimethoprim-sulfamethoxazole or combinations of pyrimethamine with clindamycin, azithromycin, or atovaquone are also used [110]. While these therapies can effectively manage acute infections, they have significant limitations. Chief among them are drug toxicity and the inability to eliminate the latent bradyzoite cysts that persist in tissues, particularly in the brain and muscles. Consequently, infections can reactivate if the immune system becomes compromised. Although rare, reports of *T. gondii* strains resistant to pyrimethamine, sulfonamides, and macrolides raise concerns about potential treatment failures [111]. These issues underscore the urgent need for new therapies that are safer and capable of targeting the chronic stages of infection.

To address these challenges, researchers are exploring various innovative therapeutic strategies. Host-directed therapies aim to enhance the immune system’s capacity to control infection by modulating cytokines or epigenetic regulators. Some repurposed drugs, such as the antidepressant sertraline, have demonstrated promising anti-toxoplasmosis effects through host-mediated pathways [112]. Sertraline treatment inhibited *T. gondii* proliferation and neuroinflammation by downregulating TNFR1/NF-κB signaling, thereby reducing parasite-induced depression-like behaviors [112]. Nanoparticle-based drug delivery systems are also under development to enhance drug uptake by infected cells and improve infiltration into hard-to-reach areas such as the brain [113,114,115]. Combining antiparasitic agents with copper-based nanoparticles in mouse models has improved cyst clearance [116]. Another cutting-edge approach involves CRISPR/Cas9 genome editing, which is being employed to identify essential parasite genes and create genetically attenuated strains that cannot form tissue cysts. These strains not only present targets for drug development but may also serve as candidates for live-attenuated vaccines. Emerging immunotherapeutic strategies include the adoptive transfer of *T. gondii* specific T cells and the use of immune checkpoint inhibitors like PD-1/PD-L1 blockade, which can reinvigorate exhausted T cells and diminish brain cyst burden in chronically infected mice [117]. Although these approaches show promising results in experimental settings, a systematic assessment of their success rates, model-specific factors, and translational challenges is crucial to evaluate their clinical feasibility and prioritize future development.

Vaccine development for human toxoplasmosis remains in the preclinical stage. Currently, no licensed vaccines are available for humans, primarily due to the parasite’s complex life cycle and the requirement for robust cell-mediated immunity, along with significant translational obstacles in moving from animal models to human application. Nevertheless, various experimental approaches are being investigated. Subunit and DNA vaccines encoding key antigens have demonstrated partial protection in animal models, especially when paired with potent adjuvants. Nanoparticle-based and mRNA vaccine platforms have shown promise by enhancing antigen presentation and immune activation. However, none of these approaches has yet achieved the sterilizing, long-term immunity necessary to fully prevent infection and cyst formation. More progress has been made in veterinary medicine. A live-attenuated vaccine (Toxovax^®^), based on the S48 strain of *T. gondii*, is successfully used in sheep to prevent abortion and reduce tissue cyst burden [118]. Nevertheless, this vaccine’s short shelf life, high production cost, and potential risk of reversion to virulence have limited its use to only a few countries. Similar efforts are underway in cats, the definitive hosts of the parasite. Experimental cat vaccines, including genetically modified strains lacking the *HAP2* gene necessary for sexual development, have been found to prevent oocyst shedding, therefore interrupting environmental transmission [119]. Other attenuated strains, like T-263, also show promise in reducing oocyst production [120]. The T-263 strain has conferred sterilizing immunity in kittens vaccinated with T-263; they shed no oocysts when challenged, whereas 84% of unvaccinated kittens shed oocysts [121]. Although no commercial cat vaccine is currently available, partly due to cost and logistical hurdles in vaccinating cat populations, these advances in animal vaccines are vital to reducing zoonotic transmission through improved veterinary control.

In summary, while current therapies for toxoplasmosis can be lifesaving, they are limited by toxicity, restricted efficacy against tissue cysts, and the threat of resistance. Emerging therapies, including host-modulating drugs, nanocarriers, gene editing, and immunotherapies, offer promising new directions for more effective treatment. Vaccine development remains an ongoing challenge but has shown encouraging results in animal models and veterinary applications. Continued translational research bridging clinical, immunological, and veterinary advances will be essential for developing safe, broadly effective tools to combat this globally prevalent parasite.

## 9. Conclusions

Toxoplasmosis poses a considerable health risk to humans, wildlife, pets, and livestock. This risk arises from its ability to persist within cells, evade and modulate innate and adaptive immune responses, and establish long-term niches in immune-privileged areas like the brain. As this review highlights, *T. gondii* not only reveals the delicate balance between protective immunity and immunopathology but also exposes exploitable vulnerabilities in host–parasite interaction.

Enhancing our understanding of how *T. gondii* manipulates PCD, subverts antigen presentation, and alters cytokine networks has profound implications for the design of host-directed therapies. Developing safe and effective vaccines, especially those capable of blocking chronic infection or transmission, remains a priority. Innovative approaches, including CRISPR-attenuated strains, nanoparticle-based platforms, and mRNA vaccines, offer new hope in tackling these challenges.

Future research should focus on underexplored aspects such as the biology of bradyzoite cysts, strain-specific immunomodulation, and immune responses in tissue-resident niches. Leveraging omics technologies, in vivo models, and systems immunology will be essential to bridge these knowledge gaps. Ultimately, *T. gondii* provides a powerful model for understanding intracellular persistence, immune modulation, and the co-evolution of host–pathogen dynamics. Insights gained from this parasite can inform therapeutic and vaccine innovation for toxoplasmosis and a range of chronic infectious diseases, advancing our broader understanding of host immunity and translational immunology.

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
