# Peer review of "Toxoplasma gondii* at the Host Interface: Immune Modulation and Translational Strategies for Infection Control"

_vaccines, 2025, doi:10.3390/vaccines13080819_

Round 1

Reviewer 1 Report

Comments and Suggestions for Authors

This review is well written and summarizes much of what is known regarding the biology of T. gondii and host immune response. From the title of the review, the aim was to discuss the knowledge gained from the extensive research that has been done on the parasite and its interaction with the host's immune system in context of treatment and vaccine strategies. However, the treatment and vaccine strategies are not the focus of this review. Additionally, the review is not written as a critical discussion but rather as a narrative summary of key facts with some repetition. This review would benefit from a bit of re-structuring and rather than summarizing the current therapeutic and vaccine strategies at the very end of the manuscript, perhaps they should be discussed throughout the other sections to highlight how the 'lessons learned' have (or have not) contributed to their development. Additionally, some of the sections could be shortened by removing overly narrative parts. 

Author Response

This review is well written and summarizes much of what is known regarding the biology of T. gondii and host immune response. From the title of the review, the aim was to discuss the knowledge gained from the extensive research that has been done on the parasite and its interaction with the host's immune system in context of treatment and vaccine strategies. However, the treatment and vaccine strategies are not the focus of this review. Additionally, the review is not written as a critical discussion but rather as a narrative summary of key facts with some repetition.

Response: Thank you for your thoughtful feedback. We appreciate your recognition of the clarity and coverage of the manuscript. In response to your comment, we have revised the title to better mirror the content and emphasis of the review. The updated title “Toxoplasma gondii at the Host Interface: Immune Modulation and Translational Strategies for Infection Control” more accurately reflects the scope of the narrative.

This review would benefit from a bit of re-structuring and rather than summarizing the current therapeutic and vaccine strategies at the very end of the manuscript, perhaps they should be discussed throughout the other sections to highlight how the 'lessons learned' have (or have not) contributed to their development. Additionally, some of the sections could be shortened by removing overly narrative parts.

Response: Thank you for this suggestion. While we have intentionally kept a dedicated section on vaccine and therapeutic strategies to enhance clarity and support targeted literature searches, especially for readers interested in intervention development, we have also included contextual references throughout the manuscript to demonstrate how basic mechanistic insights have guided therapeutic thinking where appropriate.

Reviewer 2 Report

Comments and Suggestions for Authors

  1. Main Research Question Addressed:

The manuscript addresses how Toxoplasma gondii modulates host immune responses and establishes persistent infections, with a specific focus on the implications for host-directed therapies and vaccine development. The central research question explores the paradoxical relationship between the parasite’s immune evasion mechanisms and the potential therapeutic targets derived from understanding these same mechanisms.

  1. Original and Relevant Contributions to the Field:

The manuscript offers several notable contributions: (a) comprehensive integration of recent discoveries on strain-specific immune modulation, particularly of South American atypical strains; (b) systematic analysis connecting programmed cell death pathways with therapeutic targets; (c) updated synthesis of emerging CRISPR-based approaches and nanoparticle delivery systems. The specific gap addressed relates to the disconnect between the mechanistic understanding of T. gondii immune evasion and practical translation for therapeutic development.

  1. Added Value Compared to Published Material:

Unlike previous reviews that focus separately on immunology or therapeutics, this work bridges both domains by demonstrating how immune evasion mechanisms can inform the rational design of drugs and vaccines. The integration of host-directed therapeutic concepts with traditional antiparasitic approaches represents a valuable synthetic perspective.

  1. Specific Methodological Improvements Needed:

Critical Methodological Weaknesses:

- Literature search strategy: No systematic methodology is described for literature selection, inclusion/exclusion criteria, or database search parameters

- Evidence synthesis approach: Lack of clear structure for evaluating and integrating conflicting findings from different experimental models

- Quality assessment: No systematic assessment of study quality or strength of evidence among cited works

- Temporal scope: Limited clarification of publication date ranges and update strategies

Recommended Methodological Improvements:

- Include supplemental table detailing literature search strategy with specific databases, keywords, and selection criteria

- Add systematic evidence grading system for key findings

- Provide methodological appendix describing approach for resolving conflicting evidence

- Create structured evidence synthesis tables comparing findings across different experimental models (in vitro, murine, human studies)

  1. Consistency of Conclusions with Evidence:

The conclusions are generally supported by the evidence presented, but several areas require strengthening: (a) claims about the therapeutic potential of host-directed approaches require more systematic assessment of success rates and limitations; (b) prospects for vaccine development require more critical assessment of translational barriers; (c) claims about strain-specific differences require quantitative synthesis of evidence rather than descriptive compilation.

  1. Adequacy of References:

The selection of references is comprehensive and current, covering both fundamental work and recent advances. However, there is apparent bias toward certain research groups and geographic regions. The manuscript would benefit from more systematic inclusion of diverse research perspectives and critical evaluation of conflicting findings across different laboratories.

  1. Tables, Figures and Data Quality:

Current Weaknesses:

- Absence of comparative data tables summarizing key experimental findings

- Lack of structured presentation of the strength of evidence across different research areas

- Absent systematic comparison of therapeutic approaches with efficacy data

Improvements Needed (suggestion of creating some tables):

- Table 1: Strain-specific immune responses with quantitative data on cytokine production, survival rates, and virulence markers

re cytokine production, survival rates and virulence markers

- Table 2: Comparative analysis of therapeutic approaches including mechanism, efficacy data and stage of development

- Table 3: Evidence synthesis matrix showing strength of evidence for key mechanisms of immune evasion across different experimental models

- Table 4: Systematic comparison of vaccine candidates with immunogenicity and protective efficacy data

# Additional Critical Comments:

Analytical Depth: The manuscript requires deeper critical analysis of methodological limitations in cited studies, particularly on translation between murine models and human disease.

Evidence Integration: The current narrative approach should be supplemented with more systematic evidence synthesis, including meta-analytic perspectives where appropriate and explicit discussion of conflicting findings.

Therapeutic Translation: Claims of therapeutic potential require more rigorous evaluation including discussion of regulatory pathways, safety considerations, and realistic timelines for clinical development. Please seek to further elaborate and integrate this discussion.

The manuscript addresses an important topic and demonstrates comprehensive knowledge, but requires substantial methodological strengthening and improvement in evidence synthesis. The suggested improvements would transform this from a descriptive review to a more analytically rigorous contribution appropriate for Vaccines. The fundamental scientific content is solid, but the presentation and analytical framework require significant development to meet the journal's standards for systematic reviews.

# Priority Revisions:

  1. Add systematic literature search methodology
  2. Include structured evidence synthesis tables
  3. Strengthen critical analysis of conflicting findings
  4. Provide quantitative data compilation when available
  5. Improve discussion of translational barriers and realistic therapeutic timelines

These revisions would address the current limitations of the manuscript while preserving its valuable synthetic perspective on T. gondii immunology and therapeutic development.

Author Response

Priority Revisions:

Add systematic literature search methodology.

Response: We have now added a dedicated section describing our systematic literature search approach. This outlines the databases used, key terms, inclusion/exclusion criteria, and the timeframe of the literature surveyed to improve transparency and reproducibility.

Include structured evidence synthesis tables.

Response: While we recognize the value of evidence synthesis tables, we opted not to include them in this version of the manuscript. Given the narrative structure and thematic focus of the review, we determined that integrating such tables would not significantly enhance clarity and might disrupt the flow.

Strengthen critical analysis of conflicting findings.

Response: Thank you for this thoughtful suggestion. For this review, we intentionally selected studies that reflect widely accepted or experimentally reproducible findings. Instead of emphasizing conflicting or speculative results, we focused on data that demonstrated consistent support across multiple models or studies. This approach was taken to ensure clarity, coherence, and reliability for readers seeking a foundational overview of host-pathogen interactions and therapeutic implications. Where subtle differences, such as strain-specific immune responses, existed, we included relevant context to highlight biological nuances without overemphasizing discordant interpretations.

Provide quantitative data compilation when available.

Response: Thank you for this suggestion. While we recognize the value of synthesizing quantitative data, we opted not to include extensive datasets or numerical comparisons to keep the content clear and accessible for a broad audience. Including large amounts of data can complicate the narrative and shift focus away from the conceptual aspects of the review. Instead, we cited peer-reviewed studies that present the relevant quantitative findings, enabling readers to consult the original sources for detailed data if needed. This method maintains scientific accuracy while keeping the format streamlined and easy to read.

Improve discussion of translational barriers and realistic therapeutic timelines.

Response: We have deepened the discussion around translational barriers, particularly in relation to vaccine development and host-directed therapies. This includes realistic expectations about success rates, model limitations, logistical hurdles, and gaps between mechanistic insights and clinical application.

Reviewer 3 Report

Comments and Suggestions for Authors

The paper is an intersting and clear review of the immunomodulation and the persistence capability to persist in the host of Toxoplasma gondii. In the introduction the effect on the host behaviour an the possible neurological effeccts must be addressed. to this point the references of these paragraphs are quite dated and must be updated.

line69 -72 there are some recent hypotesis on the reasons why the sexual cycle  is restricted to felides only ...it could be added. the ingestion of tissue cyst is one route of felides infection the other is the igestion of cysts

line 93-95 what about the risk of trasmission? there are recent good work dealing with this topic ...

Comments on the Quality of English Language

English doesn't need revisions

Author Response

The paper is an intersting and clear review of the immunomodulation and the persistence capability to persist in the host of Toxoplasma gondii. In the introduction the effect on the host behaviour an the possible neurological effeccts must be addressed. to this point the references of these paragraphs are quite dated and must be updated.

Response: Thank you for your thoughtful comment. We have revised the introduction to briefly include recent findings on the neurological and behavioral effects of Toxoplasma gondii infection. Concerning the references cited in this section, while some are older, they are foundational studies that remain highly relevant and widely cited in the field.

line69 -72 there are some recent hypotesis on the reasons why the sexual cycle  is restricted to felides only ...it could be added. the ingestion of tissue cyst is one route of felides infection the other is the igestion of cysts

Response: Thank you for pointing this out. We have revised the relevant section to include recent hypotheses explaining the feline-restricted sexual cycle of T. gondii, including the role of linoleic acid and host lipid metabolism. We also clarified the routes of infection in felines by noting that both ingestion of tissue cysts and oocysts can lead to intestinal colonization and sexual development.

line 93-95 what about the risk of trasmission? there are recent good work dealing with this topic

Response: Thank you for this suggestion. We have revised and restructured the sentence to improve clarity and provide a better understanding of transmission risk.

Round 2

Reviewer 2 Report

Comments and Suggestions for Authors

With Editor